# Contribution of climate change to the spatial expansion of West Nile virus in Europe

Diana Erazo [1] ✉, Luke Grant [2], Guillaume Ghisbain[1,3], Giovanni Marini [4], Felipe J. Colón-González[5], William Wint [6], Annapaola Rizzoli[4], Wim Van Bortel [7,8], Chantal B. F. Vogels [9], Nathan D. Grubaugh [9,10], Matthias Mengel[11], Katja Frieler [11], Wim Thiery [2] & Simon Dellicour [1,12] ✉

West Nile virus (WNV) is an emerging mosquito-borne pathogen in Europe where it represents a new public health threat. While climate change has been cited as a potential driver of its spatial expansion on the continent, a formal evaluation of this causal relationship is lacking. Here, we investigate the extent to which WNV spatial expansion in Europe can be attributed to climate change while accounting for other direct human influences such as land-use and human population changes. To this end, we trained ecological niche models to predict the risk of local WNV circulation leading to human cases to then unravel the isolated effect of climate change by comparing factual simulations to a counterfactual based on the same environmental changes but a counterfactual climate where long-term trends have been removed. Our findings demonstrate a notable increase in the area ecologically suitable for WNV circulation during the period 1901–2019, whereas this area remains largely unchanged in a no-climate-change counterfactual. We show that the drastic increase in the human population at risk of exposure is partly due to historical changes in population density, but that climate change has also been a critical driver behind the heightened risk of WNV circulation in Europe.

Anthropogenic climate change is having unprecedented impacts on ecosystem functions and services globally, with mounting evidence that climate-driven changes in the redistribution of biodiversity can impoverish human well-being[1]. Understanding how large-scale public health issues are modulated by anthropogenic stressors is critical as sustained changes in climate are expected to force changes in distribution and phenology of several arthropods of epidemiological importance, such as mosquitoes and ticks that can transmit pathogens[2]. The effects of altered climatic conditions on the reproduction, development, behaviour, and survival of these ectothermic organisms can have major consequences on their population dynamics at large scales,

[1]Spatial Epidemiology Lab (SpELL), Université Libre de Bruxelles, Brussels, Belgium. [2]Department of Water and Climate, Vrije Universiteit Brussel, Brussels, Belgium. [3]Laboratory of Zoology, Research Institute for Biosciences, University of Mons, Mons, Belgium. [4]Research and Innovation Centre, Fondazione Edmund Mach, San Michele all'Adige, Trento, Italy. [5]Data for Science and Health, Wellcome Trust, London, UK. [6]Environmental Research Group Oxford Ltd, Department of Biology, Mansfield Road, Oxford OX1 3SZ, UK. [7]Unit Entomology, Department of Biomedical Sciences, Institute of Tropical Medicine, Antwerp, Belgium. [8]Outbreak Research team, Department of Biomedical Sciences, Institute of Tropical Medicine, Antwerp, Belgium. [9]Department of Epidemiology of Microbial Diseases, Yale School of Public Health, New Haven, CT, USA. [10]Department of Ecology and Evolutionary Biology, Yale University, New Haven, CT, USA. [11]Department Transformation Pathways, Potsdam Institute for Climate Impact Research (PIK), Potsdam, Germany. [12]Department of Microbiology, Immunology and Transplantation, Rega Institute, Laboratory for Clinical and Epidemiological Virology, KU Leuven, Leuven, Belgium. ✉e-mail: diana.erazo.quintero@ulb.be; simon.dellicour@ulb.be

with cascading impacts on pathogen transmission to human and animal populations[3–5].

Mosquito-borne diseases such as dengue, malaria, and West Nile fever, are dynamic systems involving complex ecological interactions depending on local environmental conditions[5]. The transmission rates, range, distribution, and seasonality of their causal pathogens are notably influenced by environmental changes[6–9]. Climatic variables such as air temperature, precipitation, and relative humidity can affect the habitat suitability, distribution, bionomics, and abundance of the mosquito vectors transmitting these pathogens, as well as their host-seeking activity and biting behaviour[10–15]. Furthermore, the duration of the extrinsic incubation period (EIP) is strongly dependent on temperature[16,17]. Nonetheless, the degree to which recent observed changes in the distribution of mosquito-borne diseases can be attributed to climate change as a primary driver remains poorly understood[18].

West Nile virus (WNV) is one of the most recent and widespread emerging mosquito-borne viruses in Europe[19,20]. It is maintained in a bird-mosquito transmission cycle primarily involving mosquito species of the genus *Culex*, of which the *Culex pipiens* complex is thought to be the most common and competent vector in Europe[21–23]. Mosquitoes first get infected after biting an infectious bird and, after a temperature-dependent incubation period ranging from 2 to 14 days[24], become infectious themselves and can transmit the virus through subsequent blood feeding[25]. In this transmission cycle, mammals — particularly humans and horses — act as incidental dead-end hosts unable to re-transmit the virus to mosquitoes[25,26]. While most human and animal infections are asymptomatic, in humans around 25% of victims develop symptoms such as fever and headache, and <1% develop more severe neurological complications that can lead to death[25].

WNV has circulated in Europe since the 1950s, but it is only in 1996 that a large human outbreak with 393 human cases was detected in Romania[27]. WNV is characterised by a high genetic diversity, with West Nile virus lineage 1 (WNV-1) and West Nile virus lineage 2 (WNV-2) mainly associated with disease in humans and animals. A phylogenetic analysis has shown that six lineages have so far been detected in Europe where WNV-2 had the largest number of sequences available, accounting for 82% of all WNV sequences detected in Europe so far, and the widest diffusion since it has been found in at least 15 European countries[28]. Since its emergence on the continent, annual WNV outbreaks have been reported every summer in the Mediterranean and central Europe[29]. Since its detection in the State of New York in 1999, WNV has also invaded the North American continent[30]. Between 1999 and 2021, the USA has reported >55,000 WNV cases, of which >27,500 led to a neuroinvasive disease and >2500 to death (www.cdc.gov/westnile).

It was earlier demonstrated that the occurrence of the virus is linked to high temperatures in spring[31] and summer[32,33], droughts in summer[32,33], and warm winters[33]. In addition, high spring and summer temperatures, lower water availability, and drier winter conditions were found to be main determinants of WNV occurrence across Europe[34]. While local WNV circulation in Europe has been shown to depend on weather conditions[5,31], so far, the effect of the historical long-term changes in climate on the occurrence of (human) infections on the continent has not been quantified. An overall high sensitivity to weather conditions does not necessarily imply a strong impact of long-term climate change, as this depends on the strengths of these long-term changes in climate, the interplay across the changes in different climate variables that may amplify or cancel out, and the impact of long-term changes in other environmental and/or anthropogenic drivers. Changes in land use could indeed also noticeably impact the circulation of such vector-borne pathogens[35]. For instance, irrigated croplands and highly fragmented forests are known to favour WNV outbreaks in Europe[32,33]. Furthermore, the presence of standing water

bodies promotes the completion of the life cycle of mosquitoes and favours the sympatry between mosquito and bird populations[36,37]. Biodiversity loss can also promote transmission patterns as decreases in host community diversity could increase the vector-host encounter rate[38,39]. For example, a negative correlation has been found between bird diversity and WNV infection in vectors, at the regional scale in Missouri, and in humans, at the national scale in the USA[40]. On the other hand, some evidence also supports the assertion that avian biodiversity loss can be a contributing factor to the decline in mosquito infection rates and avian seroprevalence in Atlanta (Georgia, USA)[41].

In this context, the Working Group 2 (WG2) of the Intergovernmental Panel on Climate Change (IPCC) devoted a section to the attribution of observed changes in human, natural and managed systems to climate change in its sixth assessment report (IPCC 2022, chapter 16.2.1[42]). The framework outlined by the IPCC defines an "observed impact as the difference between the observed state of a natural, human or managed system and a counterfactual baseline that characterises the system's state in the absence of changes in the climate-related systems", where climate-related systems mean the climate-system itself including the ocean and the cryosphere (e.g., changes in sea level rise) as physical or chemical systems that are not relevant in this study. The IPCC then states that the "difference between the observed and the counterfactual baseline state is considered the change in the natural, human or managed system that is attributed to the changes in the climate-related systems (impact attribution)". "Changes in climate-related systems" explicitly mean "any observed long-term change" no matter whether such a trend is induced by anthropogenic climate forcing or not[43]. The counterfactual impact baseline cannot be observed and thus needs to be simulated by an impact model. A precondition for impact attribution is that the impact model explains the observed phenomenon under consideration reasonably well given its drivers.

Here, we investigate the contribution of climate change (observed long-term trend in climate) to WNV spatial expansion in Europe building on the IPCC framework. We use an impact indicator of WNV infection risk, assess its performance for the historical period and compare it to a counterfactual impact baseline. We use four observationally-based reanalysis climate datasets and their counterfactuals that were recently made available through the Inter-Sectoral Impact Model Intercomparison Project (ISIMIP). ISIMIP is dedicated to fostering impact attribution following the definition of the IPCC WG2 in an international modelling effort in its currently running ISIMIP3a phase. Specifically, the counterfactual climate data are obtained by detrending the observational (factual) climate data: the counterfactuals approximate a "no-climate change" climate through the removal of the long-term trend related to global mean temperature change from the factual reanalysis datasets[44]. The resulting time series thus consist of stationary climate data obtained from observational daily data when removing the long-term trend while preserving the internal day-to-day variability[44]. We exploit the newly available factual and counterfactual historical datasets to assess whether there is an increased ecological suitability for WNV in Europe, and if this increase can be attributed to climate change.

## Results

We performed ecological niche modelling with a boosted regression tree (BRT) approach to estimate the WNV ecological suitability, a metric ranging from 0 to 1 that can be interpreted as a measure of the risk of WNV circulation given local environmental conditions. We conducted the analyses at the "nomenclature of territorial units for statistics" (NUTS) level 3, i.e., administrative polygons corresponding to the third administrative level in European countries. The input data were twofold: 13 years of confirmed WNV human infections across the European continent for the period (2007–2019; Fig. 1), as well as

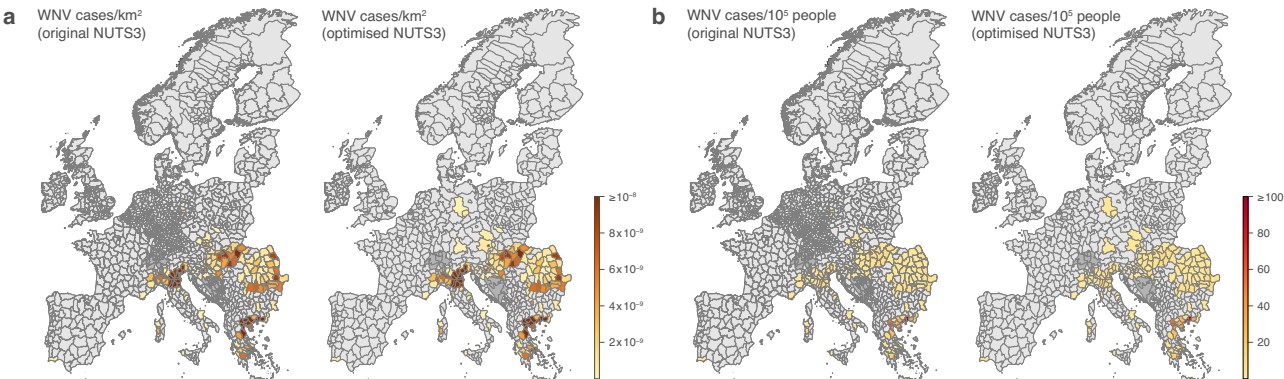

**Fig. 1 | Distribution of WNV cases reported in the ECDC database until 2019.** We here report the density of both "probable" and "confirmed" cases in each administrative area for which there was at least one confirmed case before 2020. Specifically, we report the number of WNV cases by aggregating both "probable" and "confirmed" per km² (**a**) or per 100,000 people (**b**), either considering the NUTS3 or optimised NUTS3 administrative polygons. Darker grey areas correspond to Switzerland and Bosnia & Herzegovina, two countries that are not included in the ECDC WNV surveillance. The administrative areas of those countries were not considered when training the ecological niche models. The United Kingdom is also not included in the ECDC WNV surveillance but was considered when training the models as the country has not reported any WNV case so far (www.nhs.uk).

climatic, land-use, and human population data for a corresponding period (2000–2019; Fig. S1). We trained our models on such present-day data retrieved from four ISIMIP3a reanalysis datasets of historical climate change (GSWP3-W5E5, 20CRv3, 20CRv3-ERA5, and 20CRv3-W5E5). We subsequently estimated the areas ecologically suitable for local WNV circulation leading to human cases since the beginning of the 20th-century considering either the historical climate or its respective counterfactual. Since the simulated historical WNV ecological suitability notably differs among the four ISIMIP3a reanalysis datasets used to train the models (see below), we did not average their outcome and independently reported the simulations based on each dataset considered, which allowed pointing and discussing their differences.

With Area Under the receiver operating characteristic Curve (AUC) and prevalence-pseudoabsence-calibrated Sørensen's Index ($SI_{ppc}$) metrics all higher than 0.8 (Table S1, Fig. S2; see also the "Methods" section for further detail on those metrics), our ecological niche models demonstrate a good prediction performance when trained on the ECDC data. Those ecological niche models also allowed us to compute the relative influence (RI) of each environmental factor in the models (Table S2). For the GSWP3-W5E5 factual climate dataset, near-surface air temperature in summer demonstrated the highest relative influence (RI = 16.3%), followed by air temperature in winter (RI = 10.5%), relative humidity in winter (RI = 10.1%), precipitation in summer (RI = 8.5%), and relative humidity in fall (RI = 7.5%; Table S2). As for the land-use factors in the ecological niche models, the proportion of managed pastures and rangeland presented the highest relative influence (RI = 7.1%), followed by secondary non-forested areas (RI = 4.8%), and croplands (RI = 4.8%). These results are globally consistent with those obtained from training the ecological model on three alternative climate reanalyses (Table S2).

To assess the relationship between each environmental factor and the estimated ecological suitability values, we plotted response curves showing how WNV ecological suitability varies with one specific environmental factor (Fig. S3). For instance, air temperature and precipitation in summer, as well as croplands, display a clear positive association with WNV ecological suitability, with summer air temperature above ~20 °C, summer precipitation above ~2.2 kg/m²/day, and cropland density >20% being associated with a notable increase in WNV ecological suitability (Fig. S3). Conversely, environmental variables, such as air temperature in winter, relative humidity in winter, and density of managed pastures as well as rangeland, all presented a negative association with WNV ecological suitability: the estimated

ecological suitability displays a notable decrease for winter air temperature > 4–5 °C, winter relative humidity >83–84% and managed pasture/rangeland density >10% (Fig. S3).

To investigate the impact of climate change on the spatial expansion of WNV across the continent, we compare past changes in WNV ecological suitability based on ecological niche model simulations forced by the factual and counterfactual historical climate, respectively. We first considered the simulations based on the GSWP3-W5E5 reanalysis dataset, which is particularly aligned with real-world conditions for the recent years coinciding with the time window of WNV case data obtained from the ECDC and on which our ecological niche models were trained. W5E5 is considered the potentially closest approximation to reality as it is based on the latest version of the European Reanalysis (ERA5)[45] that was further corrected by observational data based on the WATCH Forcing Data methodology[46]. To generate the counterfactuals, i.e., to construct a dataset that described a counterfactual world without long-term changes in climate since 1901, the W5E5 data had to be expanded backwards in time. For this extension, we first used version 1.09 of the Global Soil Wetness Project phase 3 (GSWP3) dataset[47], bias-adjusted to W5E5 v2.0 in order to reduce discontinuities at the 1978–1979 transition[44]. As some variables in GSWP3 show discontinuities at every turn of the month that have been induced by a month-by-month bias adjustment to the underlying raw data (20CRv2)[48], we additionally considered a backward-extension based on the Twentieth Century Reanalysis version 3 (20CRv3)[49,50], interpolated to 0.5° and then bias-adjusted to W5E5 v2.0. Notably, 20CRv3-W5E5 data remain continuous at every turn of the month thanks to the application of ISIMIP3BASD v2.5 in running-window mode. Thus, 20CRv3-W5E5 reanalysis dataset can be considered an update of GSWP3-W5E5. 20CRv3-ERA5 has then been introduced to allow for testing the sensitivity of the results to potential trend and variability artefacts in W5E5 that are related to the climatological infilling procedures used to deal with gaps in the station observations employed for the bias adjustment of ERA5 for the production of WFDE5 (for a detailed description of this caveat see https://data.isimip.org/caveats/20/). Finally, we have also considered the "raw" 20CRv3 data interpolated to 0.5° but not bias-adjusted to any other dataset. This latter dataset is included since it was generated with only one method and did not need to be combined with another dataset to fully cover the 20th-century. Overall, considering all four reanalysis datasets (that have all been introduced by Mengel and colleagues[44]) allows us to investigate the robustness of our findings to the choice of the reanalysis dataset.

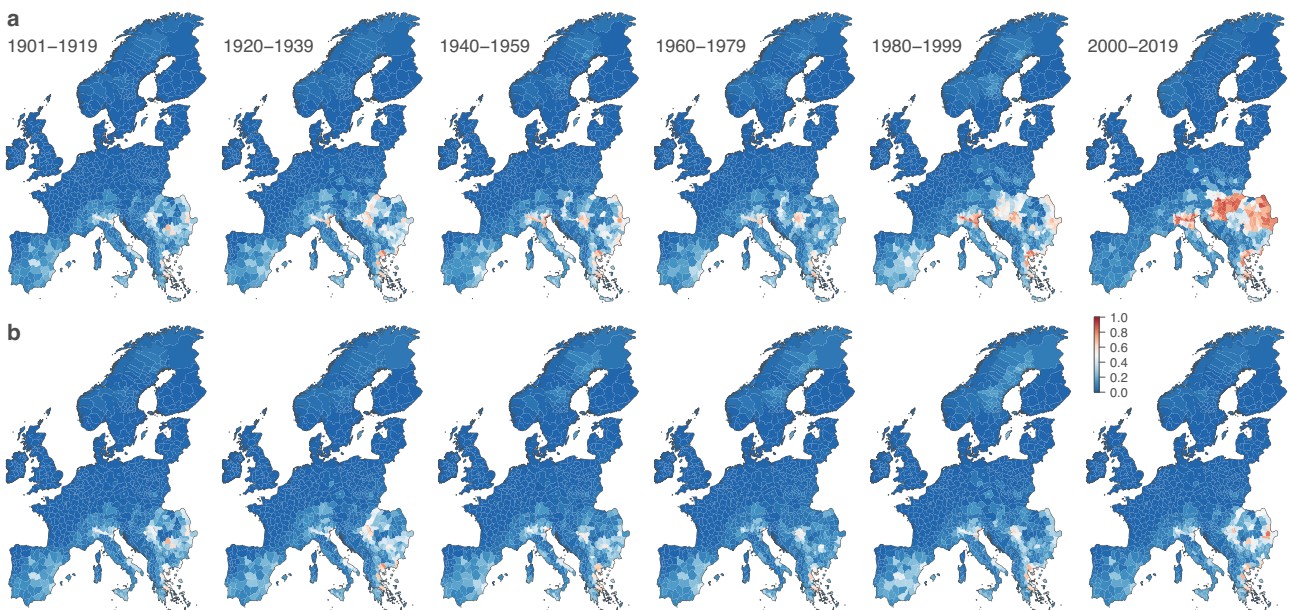

**Fig. 2 | Changes in the ecological suitability of West Nile virus (WNV) in Europe.** Past and present ecological suitability estimated for each administrative unit are based on both the reconstructions of the historical climate (**a**) and a counterfactual baseline (**b**). Ecological suitability values are averaged over the estimates of ten independent BRT models trained on present-day data retrieved from the ISIMIP3a reanalysis dataset GSWP3-W5E5. See Figs. S4–S6 for the estimates of three other ISIMIP3a reanalysis datasets (20CRv3, 20CRv3-ERA5, and 20CRv3-W5E5).

Under factual climate conditions, model simulations based on the GSWP3-W5E5 reanalysis dataset show clear increases in WNV ecological suitability in European regions such as northern Italy, the Carpathian mountains, the lowlands in Hungary and eastern Romania, and the Aegean region in eastern Greece (Fig. 2a). The increase in WNV ecological suitability is particularly marked from the 1980s. The noticeable increase is not present in the reconstructions driven by the counterfactual GSWP3-W5E5 data: in absence of historical climate change, the estimated WNV ecological suitability map remains similar throughout the 1901–2019 period (Fig. 2b). In other words, we do not find a clear increase in the local risk of WNV circulation when non-climatic environmental factors change as observed over the past century, but climate change is absent. These results indicate that climate change contributed to the escalation of the risk associated with West Nile virus circulation in Europe.

While we can also highlight two different trends when comparing the past evolution of the WNV ecological suitability projected on historical and counterfactual data from the 20CRv3-W5E5 reanalysis dataset (Fig. S4) and, to some extent, the 20CRv3-ERA5 dataset (Fig. S5), this is however not the case with the 20CRv3 dataset (Fig. S6). Yet, as the 20CRv3 reanalysis only assimilates surface pressure in combination with prescribed sea surface temperatures and sea ice concentrations, it can be considered the most uncertain product for present-day conditions. Among the meteorological variables with the highest RI in the ecological niche models, in 20CRv3, relative humidity changes the least from early 20th-century conditions (Fig. S7).

For the three other reanalysis datasets, relative humidity in winter is associated with an averaged RI above or close to 10% and decreases across the continent during the last century. Considering the negative relationship estimated between winter relative humidity and WNV ecological suitability, the former emerges as a pivotal environmental factor potentially driving local increments in the latter. With an averaged RI higher than 15% no matter the reanalysis dataset (Table S2) and associated response curves highlighting a clear positive association with WNV ecological suitability (Fig. S3), air temperature in summer is here identified as another predominant factor: an increase in summer

air temperature across the continent (Fig. S8) is indeed concomitant with the increased risk of WNV circulation.

We then compare the evolution of the estimated population at risk of exposure to WNV under factual and counterfactual historical climate. We estimate the total number of people at risk of exposure across the entire study area while considering two different threshold values of ecological suitability above which we consider a risk of local WNV circulation (0.1 and 0.5). As expected, both the historical and counterfactual climate are associated with similar values at the beginning of the last century. However, the estimates then diverge with time, as the historical population at risk of exposure almost doubles that of the counterfactual by the onset of the 21st century in the GSWP3-W5E5 reanalysis dataset and a threshold ecological suitability value of 0.1 (Fig. 3). When considering a threshold value of 0.5 for the estimates derived from the same reanalysis dataset, the historical population at risk of exposure is almost multiplied by six when compared to that of the counterfactual at the present-day (Fig. 3). In the counterfactual baseline case, the increase in the number of people at risk of exposure is thus largely due to the increase of the European population from 1901 to 2020. The estimates we obtained based on the three other reanalysis datasets are however different: although comparable patterns can be observed, the differences between the estimated historical and counterfactual populations at risk of exposure is less marked (Fig. 3).

## Discussion

European public health authorities have reported autochthonous cases of WNV in 2011[51], which has been suggested to be mainly driven by favourable environmental conditions for the establishment of the virus, such as warm spring and summer seasons[32–34,52], heavy rains[33], and river floods[52]. This study is a first attempt to formally evaluate the attribution of climate change to the enhanced circulation of West Nile virus in Europe. Our results indicate that the development of the current hotspots of WNV circulation in Europe can be to a large extent attributed to climate change. These findings are in line with previous studies showing that increases in summer air temperature[32] and precipitation[33] represent pivotal drivers of WNV circulation. Our

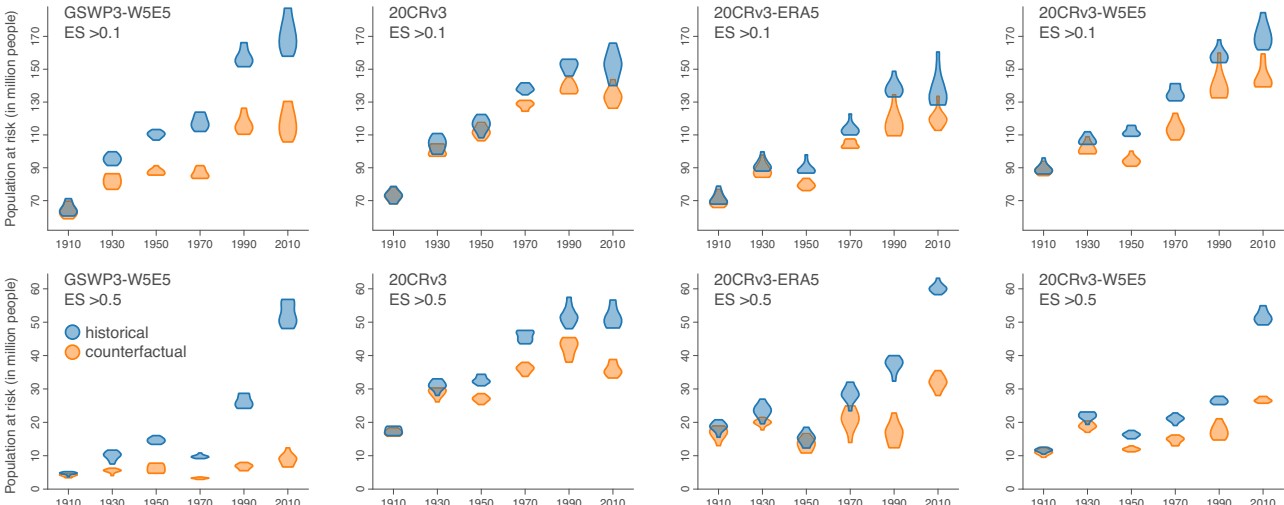

**Fig. 3 | Changes in the population at risk of exposure to West Nile virus (WNV) in Europe.** Past changes in the population at risk of exposure to WNV have been estimated for both a counterfactual baseline and the observed historical climate data retrieved from four ISIMIP3a reanalysis datasets (GSWP3-W5E5, 20CRv3, 20CRv3-ERA5, and 20CRv3-W5E5) and while considering two different thresholds of ecological suitability (ES) above which an area was considered at risk (0.1 and 0.5).

assessment indicates that Europe experienced a notable increase in WNV ecological suitability from the 1980s, which coincides with a rapid warming during this time in Europe (https://showyourstripes.info/c/europe/all/) as well as the establishment of WNV hotspots in Romania from 1996[27,52], in Italy from 2008[53], and in Greece from 2010[54].

Although anomalous higher temperature in spring was previously identified as a reliable early warning predictor for WNV human cases[31,34], we here find that the relative contributions of air temperature in summer and winter to the ecological suitability of WNV were higher than the one of spring temperatures. This discrepancy could be due to the definition of the response variable of the model: while other studies define it as incidence[31,55], we define it as a binary presence/absence variable, which is a different way to account for WNV circulation (see below for a motivation of that choice). Moreover, it has been shown that factors associated with WNV incidence in the USA on a 10-year scale could be different from those associated with inter-annual differences[56]. Irrigated croplands and fragmented forests have also been pointed as factors that might favour WNV outbreaks in Europe[32,33]. Our results also show that the ecological suitability of WNV could increase with cropland density, although managed pastures and rangelands could also play a role.

Our study presents necessary analytical compromises. First, it was not feasible to incorporate important biotic factors such as mosquito and bird diversity/abundance in our ecological niche modelling analyses. While some distribution data are available for the *Culex* species involved in WNV transmission cycle, there is no equivalent data encompassing the past century, which would prevent us to re-project the WNV ecological suitability from 1901 onward. One way to circumvent the first issue would be to train and project in the past ecological niche models for those *Culex* species, but such ecological niche models would themselves be based on the same or a very similar set of present-day environmental factors to the one also included in the set of variables used to train WNV ecological niche models, which would thus lead to a circularity issue also making it more difficult to interpret the results (and in particular the relative influence value computed for each variable). Besides, we here aim to estimate the risk of local WNV circulation *given* local environmental conditions, which implicitly involves a minimum and here unestimated level of mosquito vector abundance. Regarding the bird data, in addition to the circularity issue mentioned above, we have also so far been unable to find a definitive

consensus list of bird species involved, and obtaining high-resolution species-specific distribution data has proved problematic anyway.

Second, considering a higher number of more detailed land-use variables would have been advantageous. Yet, the main objective of the study is not to provide the most accurate and precise WNV niche modelling, which has been investigated in previous studies[33,57,58] and would have required consideration of additional environmental predictors such as more detailed land-use categories. For example, land-use diversity, where mosquito-bird encounters could take place and amplify the viral circulation, could be a determinant for assessing the WNV risk. However, given that our analytical approach depends on the available ISIMIP environmental data, we were unable to include such more detailed land-use categories.

Third, we model WNV ecological suitability on the basis of presence/absence data, which gives the same weight to all administrative units with at least one confirmed non-imported human case, irrespective of the total number of cases reported for each area. On the other hand, it precisely prevents treating absolute number of cases as a reliable proxy for WNV prevalence, which would correspond to a relatively strong assumption given spatial heterogeneity in surveillance effort. Indeed, the case data to which we have access is reported rather than the result to more reliable serological surveys. Furthermore, considering such a binary response variable rather than an incidence estimated from reported cases also prevents us from having to make the assumption that the surveillance effort was homogeneous during the considered time period, which is likely not true given that local reporting rates could increase over time with awareness/knowledge of WNV.

Fourth, we here attribute WNV spatial expansion in Europe to climate change (irrespective of the cause) following the IPCC AR6 WG2 framework[42] rather than to anthropogenic climate forcings, as our counterfactuals consist of detrended reanalyses[44]. Attribution to anthropogenic climate forcings is indirect here given the well-established anthropogenic influence on climate warming in Europe[59]. Finally, to match with the ISIMIP data available until 2019, our models have been trained on occurrence data reported until the same year, thus discarding from the models training more recent WNV occurrence records, e.g., reported in the Netherlands or in southern Spain. Interestingly, our ecological niche models do not highlight this area as particularly suitable for local WNV circulation, thus calling for updated ecological niche modelling for this virus in the near future.

Here we exploit newly available counterfactual climate data from ISIMIP to show that climate change is directly involved in the increase in local circulation of WNV in Europe. Our simulations of the historical period show no increase in areas ecologically suitable for WNV in the counterfactual simulations for three out of four reanalysis datasets. Our results thus point towards a significant responsibility of climate change in the establishment of WNV in the south-eastern part of the continent. In particular, we identify that current WNV hotspots in Europe are most likely to be attributed to climate change. With climate change emerging as a critical public health challenge, future work should explore the evolution of infectious disease distributions under different scenarios of future climate change to inform surveillance and intervention strategies[60,61]. In that perspective, this work provides an example of how climate data could be effectively used in an epidemiological context by estimating the past and present-day ecological suitability of the virus, filling another analytical gap between climate science and spatial epidemiology.

## Methods

### Data acquisition

WNV human infection records aggregated at the NUTS level 3 (NUTS3) from 2007 to 2019 were retrieved and curated from the European Surveillance System (TESSy) database of the European Center for Disease Prevention and Control (ECDC; Fig. 1). Climate, land-use, and population data were retrieved from the Inter-Sectoral Impact Model Intercomparison Project phase 3 (ISIMIP3, https://www.isimip.org/protocol/3/). ISIMIP prescribes protocols and background datasets for modelling the impacts of climate change in various systems sensitive to climate and human management. In ISIMIP3a, modellers run historical impact simulations with reanalysis datasets, which are global reconstructions of the historical climate. Importantly, these datasets are provided additionally in a counterfactual version without the climate change signal to enable model evaluation and attribution of the impacts of climate change. In synthesis, counterfactual time series thus consisted of stationary climate data obtained from observational daily data after having removed the long-term trend while preserving the internal day-to-day variability[44]. Climate information used in this study consists of daily gridded near-surface air temperature, surface precipitation, and relative humidity (Fig. S1). Climate data, i.e., temperature, precipitation, and relative humidity, were aggregated by season along 20-year mean periods: winter (December, January, February), spring (March, April, May), summer (June, July, August), and autumn (September, October, November). Land-use data were retrieved from the Land Use Harmonisation project (version 2; LUH2) providing historical and projected land-use states[62]. Land-use data included six land cover categories[63]: primary forest areas, primary non-forest areas, secondary forest areas, secondary non-forest areas, croplands, and rangelands/pastures (Fig. S1). Gridded human population data[62] was $\log_{10}$-transformed and divided by polygon area (km$^2$).

### Optimised polygon map

The sizes of NUTS3 polygons vary considerably among European countries. For example, NUTS3 level polygons for Germany are relatively small and similar in size to NUTS4 of other countries. We therefore used an optimised NUTS map in which levels were chosen to homogenise as much as possible the polygon unit size (Fig. 1). This standard shapefile was developed for the European network for medical and veterinary entomology (VectorNet), a project led by the ECDC and the European Food Safety Authority (EFSA) that aims to contribute to improving preparedness and response for vector-borne diseases following a one health approach[64].

### Ecological niche modelling

We implemented a BRT approach to train the different ecological niche models for WNV for the 2000–2019 period. BRT is a machine learning methodology that can be used to generate a collection of sequentially fitted regression trees optimising the predictive probability of occurrence given local environmental conditions[65]. Such a predictive probability can be interpreted as a measure of ecological suitability, a value that ranges between 0 and 1. The interest of a BRT approach lies in its ability to model complex non-linear relationships between the response and various predictor variables[66]. Additionally, a BRT approach does not require prior data transformation or elimination of outliers. Of note, it has been shown that the BRT methodology has a superior predictive performance compared to alternative modelling methods[65]. For implementing a Bernoulli BRT approach, both presence and absence data are required, in the context of this study, optimised NUTS3 administrative areas presenting one or more confirmed non-imported cases were considered as "presence" locations, and the remaining administrative areas with zero confirmed non-imported cases were treated as "pseudo-absence" locations. As stated above, we preferred using presence/absence rather than incidence data to account for surveillance heterogeneity across the study area and to thus avoid treating absolute number of cases as a reliable proxy for WNV prevalence. The response variable considered here is thus the detection or not of at least one human case during the period 2007–2019 and that is not labelled in the ECDC database as an "imported case".

We used the BRT algorithm implemented in the R package "dismo" (version 1.3–9)[67]. To account for spatial autocorrelation and avoid model overfitting, we implemented a spatial instead of a standard cross-validation procedure, the latter being known to frequently overestimate the ability of the model to make reliable predictions when occurrence data are spatially auto-correlated[68]. Specifically, we applied the spatial cross-validation procedure based on the blocks generation, described by Valavi and colleagues and implemented in the R package "blockCV" (version 3.1-1)[69]. In summary, data on WNV human infection records (Fig. 1) was first transformed as a presence/absence dataset, which was then divided in five spatial folds following the blocks generation method. BRT models were trained using the following parameter values: a tree complexity of 5, a learning rate of 0.005, and a step size of 10. We assessed the sensitivity of the predictive performance of the BRT models to the specification of the tree complexity and learning rate parameters values. As summarised in Table S3, our tests confirmed that their performance did not seem impacted by the choice of these parameter values.

The predictive performance of each BRT model was first assessed by estimating the area under the receiver operating characteristic (ROC) curve, also referred to as "area under the curve" (AUC). Because the use of the AUC metric was repeatedly criticised in previous works due to its dependence on prevalence (i.e., the proportion of recorded sites where a given species is present)[70–73], we further assessed the predictive performance of our ecological niche models by computing a prevalence-pseudoabsence-calibrated Sørensen's index (SI$_{ppc}$) defined as follows[73–75]:

$$SI_{ppc} = (2 \times TP)/(2 \times TP + x \times FP_{pa} + FN) \qquad (1)$$

where

$$x = (P/A) \times ((1 - prev_{sp})/prev_{sp}) \text{ and } prev_{sp} = P/(P + A) \qquad (2)$$

with TP corresponding to the number of true positives, FP$_{pa}$ to the number of false positives associated with sampled pseudo-absence points, FN to the number of false negatives, P to the number of presence points, and A to the number of pseudo-absence points. This index has its lower limit at 0 and its upper limit at 1 (interpreted as a maximal predictive performance). Because the computation of this index requires binary presence-absence data but ecological niche models instead return ecological suitability values ranging from 0 to 1,

we performed an optimisation procedure similar to one adopted by Li & Guo[76] by varying the threshold values in the range [0, 1] with a 0.01 step increment, and eventually selecting the threshold value maximising the $SI_{ppc}$ (see also the work of Ghisbain and colleagues for a similar approach[73]). We computed a $SI_{ppc}$ for each of the ten independent ecological niche models (Table S1). As detailed in Table S1, all averaged $SI_{ppc}$ values are higher than 0.80 (>0.83). For each trained ecological niche model, we also report the evolution of the $SI_{ppc}$ according to the ecological suitability threshold value ranging from 0 to 1 (Fig. S2).

We trained ten independent replicate BRT models on present-day data retrieved from four ISIMIP3a climate reanalysis datasets: GSWP3-W5E5, 20CRv3, 20CRv3-ERA5, and 20CRv3-W5E5[77]. The GSWP3-W5E5 (Global Soil Wetness Project Phase 3 - W5E5) dataset is a land surface reanalysis dataset, combination of GSWP3 for 1979–2019[47] with W5E5 for 1901–1978[46,78]. To minimise discontinuities at the 1978/1979 transition, GSWP3 data were homogenised with W5E5 data for 1901–1978 using the ISIMIP2BASD v2.3 bias adjustment method[79]. The 20CRv3 (The Twentieth Century Reanalysis version 3) dataset is an atmospheric reanalysis dataset and covers daily data for the period 1901–2015[80]. The 20CRv3-ERA5 is the combination of ERA5 for 1979–2021 with 20CRv3 homogenised to ERA5 for 1901–1978. Finally, the 20CRv3-W5E5 dataset is the combination of W5E5 v2.0 for 1979–2019 with 20CRv3 homogenised to W5E5 for 1901–1978[49,50]. The homogenisation for 20CRv3-ERA5 and 20CRv3-W5E5 was done in the same way as for the GSWP3-W5E5 dataset[79]. As detailed above, results for BRT models trained on GSWP3-W5E5 were here emphasised over other reanalyses because this can be considered a priority forcing dataset in ISIMIP3a. Bias adjustments to ISIMIP3b are indeed based on GSWP3-W5E5, and Mengel and colleagues[44] also evaluated the GSWP3-W5E5 factual and counterfactual datasets in their review of the ATTRICI (ATTRIbuting Climate Impacts) method for generating counterfactual climate data.

We projected each trained ecological niche model on past (1901–1999) and present (2000–2019) environmental conditions considering both a counterfactual baseline and the observed historical climate. The comparison between observed historical and counterfactual projections allowed us to discuss if and to what extent the increase in WNV ecologically suitable areas across the continent could be attributed to climate change. In particular, we estimated and compared the changes in the human population at risk of exposure to WNV in Europe for both a counterfactual baseline and the observed historical climate data retrieved from the four ISIMIP3a reanalysis datasets.

We conducted additional analyses to investigate the robustness and sensitivity of our results to the sampling intensity of pseudo-absences across the study area. While the main analyses are based on pseudo-absences sampled across 100% of the administrative areas in which no presence has been confirmed, we additionally considered alternative datasets of pseudo-absences corresponding to only 50% and 75% of the administrative areas not associated with a presence record. In both cases, we re-trained 100 ecological niche models each based on a random selection of 50% or 75% of the original pseudo-absences considered in the main analyses. Based on these newly generated models, we then re-estimated the changes in the population at risk of exposure to WNV in Europe. Overall, our results confirm that the inferred trends remain highly consistent with the results obtained when considering all potential pseudo-absences (Fig. S9). All analyses were performed using R (R Statistical Software version 4.02, R Foundation for Statistical Computing, https://www.r-project.org/).

### Reporting summary
Further information on research design is available in the Nature Portfolio Reporting Summary linked to this article.

## Data availability
All ISIMIP3a data used are publicly available on https://data.isimip.org/. Data on WNV human infections can be obtained on request from The European Surveillance System (TESSy) at the European Center for Disease Prevention and Control (ECDC) on https://www.ecdc.europa.eu/en/publications-data/european-surveillance-system-tessy. Restrictions apply to the availability of the ECDC data, which were used under license for the current study, and so are not publicly available.

## Code availability
R scripts related to the ecological niche modelling are all available at https://github.com/sdellicour/wnv_enm_europe (https://doi.org/10.5281/zenodo.3764823). All information to reproduce figures is available in the github repository.

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

## Acknowledgements

The authors acknowledge the European Center for Disease Prevention and Control (ECDC) for providing data from The European Surveillance System (TESSy) on WNV human infections. Disclaimer: The views and opinions of the authors expressed herein do not necessarily state or reflect those of ECDC. The accuracy of the authors' statistical analysis and the findings they report are not the responsibility of ECDC. ECDC is not responsible for conclusions or opinions drawn from the data provided. ECDC is not responsible for the correctness of the data and for data management, data merging and data collation after provision of the data. ECDC shall not be held liable for improper or incorrect use of the data. DE acknowledges support from the European Union's Horizon 2020 research and innovation programme under the Marie Skłodowska Curie grant agreement no. 801505. GG acknowledges support from the *Fonds National de la Recherche Scientifique* (F.R.S.-FNRS, Belgium). CBFV acknowledges support from CTSA Grant Number UL1 TR001863 from the National Center for Advancing Translational Science (NCATS), a component of the National Institutes of Health (NIH). The contents of this publication are solely the responsibility of the authors and do not necessarily represent the official views of NIH. SD acknowledges support from the F.R.S.-FNRS (grant no. F.4515.22) and the Research Foundation - Flanders (FWO, *Fonds voor Wetenschappelijk Onderzoek - Vlaanderen*, grant no. G098321N). The outbreak research team of the Institute of Tropical Medicine (ITM) is supported by the Department of Economy, Science and Innovation of the Flemish government, Belgium. This project has received funding from the European Union's Horizon 2020 research and innovation programme under grant agreement no. 874850 (MOOD project) and is catalogued as MOOD 077. The contents of this publication are the sole responsibility of the authors and don't necessarily reflect the views of the European Commission. For the production and coordination of the Inter-Sectoral Impact Model Intercomparison Project (ISIMIP; www.isimip.org) input data and impact model output, we are grateful to the modelling groups, the ISIMIP sector coordinators, and the ISIMIP cross-sectoral science team. Part of the computational resources and services used in this work were provided by the VSC (Flemish Supercomputer Center), funded by the FWO and the Flemish Government, department EWI.

## Author contributions

Study design: D.E., W.T., and S.D. Data collection and curation: D.E. and L.G. Data analysis: D.E., L.G., M.M., K.T., W.T., and S.D. Results interpretation and discussion: D.E., L.G., G.G., G.M., F.J.C-G., W.W., A.R., W.V.B., C.B.F.V., N.D.G., M.M., K.F., W.T., and S.D. Writing - original draft preparation: D.E. and S.D. Writing - review and editing: D.E., L.G., G.G., G.M., F.J.C-G., W.W., A.R., W.V.B., C.B.F.V., N.D.G., M.M., K.F., W.T., and S.D. Supervision: S.D.

## Competing interests

The authors declare no competing interests.
