## [Peer Review File · Nature Communications]

Contribution of climate change to the spatial expansion of West Nile virus in EuropeREVIEWER COMMENTS

Reviewer #1 (Remarks to the Author):

Summary: The authors present an analysis of the ecological niche of West Nile virus (WNV) in Europe and track changes in WNV suitability under historic climate projections. A state-of-the-art niche modeling approach (boosted regression trees) is used to associate climatic and land cover variables with the occurrence of human WNV cases as reported in The European Surveillance System (TESSy) database of the ECDC. The main research question addressed by this paper is to which extent the recent emergence of WNV in Europe can be attributed to climate change. The authors tackle an interesting and relevant research question. Overall, the work offers interesting insights into the climatic and land use drivers of WNV in Europe (most of which are known from previous literature), provides evidence for impact attribution, and uses advanced and robust methods. Given extensive previously published work on WNV ecological niche mapping already published (1,2) with similar and alternative methods as well as historical and future projections of WNV risk in Europe based on these (see below), the work does only provide minor novel insights and evidence for the field.

Main Strength: The authors use recently published historical and counterfactual climate datasets provided through ISMIP. A strength of the study is that they provide a direct comparison between model projections under the historical vs "no climate change" counterfactual scenarios. Moreover, the use and presentation of their results under 4 different climate models gives a good indication of sensitivity of their results to uncertainty in the climate datasets. The analysis of relative importance of different predictors, which is essential when using "black-box"-type machine learning models, is well presented.

Major Weakness: The added value and novelty of the study is not clear given other groups have previously published ecological niche modeling studies for WNV in Europe describing similar patterns and presenting similar findings (1). The most recent European Lancet Countdown report also uses the model by Farooq et al. to project historical trends in WNV risk for Europe (2). Although the Lancet Countdown indicator does not the climate change pre-industrial control counterfactual modelled scenarios in the previous work, it still provides a strong indication of to which extent changes in WNV risk can be attributed to climate change. The present work supports previous findings by approaching the topic from a slightly different angle but does not lead to major novel insights. Most of the relationships derived from the analysis of importance of the different predictors are known from previous studies in and outside of Europe. Although the new study provides some slightly alternative ranking of variable importance (such as highlighting the importance of humidity and summer vs spring temperatures (see Line 228-230) that further refine knowledge about the ecological niche of WNV in Europe.

[1] Farooq, Z., Rocklöv, J., Wallin, J., Abiri, N., Sewe, M. O., Sjödin, H., & Semenza, J. C. (2022). Artificial intelligence to predict West Nile virus outbreaks with eco-climatic drivers. *The Lancet Regional Health–Europe*, 17.

[2] van Daalen, K. R., Romanello, M., Rocklöv, J., Semenza, J. C., Tonne, C., Markandya, A., ... & Lowe, R. (2022). The 2022 Europe report of the Lancet Countdown on health and climate change: towards a climate resilient future. *The Lancet Public Health*, 7(11), e942-e965.

Minor comments:

- I am wondering how hyperparameter tuning was done since it is not reported in the text. Boosted regression trees come with several hyperparameters that need to be adjusted to construct a model with good performance. Usually this is done by either evaluating the performance of the model for different hyperparameter settings on a separate validation set or by using cross-validation, whereas evaluation of the final model performance is done on a separate out-of-sample test dataset (which should not be seen by the model during training or during validation). Here the authors construct ten different tree models using cross-validation for evaluating the performance of their tree model. The important topic of hyperparameter tuning and evaluation of their model on

a separate test dataset is missing.

- The manuscript is missing an explanation for why GSWP3-W5E5 was chosen as a reference climate dataset. What makes this climate model a reasonable choice for reporting/visualization results in the main text compared to the other three which are mainly reported in the supplementary material?
- The discussion on the relationship between avian diversity and WNV risk (Line 87-90) does not reflect the complete state of evidence. The authors suggest the presence of a dilution effect based on one reference. However, WNV studies have detected both dilution and amplification effects potentially because WNV transmission potential depends on avian community composition.
- When plotting the response curves shown in Figure S2, I suggest the authors to report how values for the other input variables of the tree models were set. Is the shape of the response curves sensitive to the values selected for the other variables?

Reviewer #2 (Remarks to the Author):

The manuscript investigates the potential role of climate change on human WNV risk in Europe. They use BRT models on historical presence/absence data together with multiple environmental variables to then predict WNV ecological suitability based on climatic projections assuming CG or lack of (counterfactual). They found that the area ecologically suitable for WNV under a CG scenario (assumed current status) considerably increased during the last century compared to the no CG counterfactual.

The manuscript is very topical and well presented with clear (and pretty) figures. My main comments are about the lack of clarity in the objective/rationale/novelty and the description of the methods, which raises questions over the conclusions. I detail these and other minor comments below.

Main comments:

Objective/motivation: The objective of the manuscript is not clear. Investigation of the contribution of CG to WNV emergence building on the IPCC framework (L103-111) is vague and confusing. In addition, 'emergence' seems to be used as a synonym to 'risk' or 'spatial expansion' as WNV has been in Europe for a while. What is the ISIMIP and the implications for WNV ecological suitability. How does the IPCC framework fit in all this?

Also, the approach does not consider vectors (nor they are ever discussed) so the assumption is that the presence of human cases implies presence of vectors. I agree this is necessarily true for VBDs, my concern is that it also assumes a linear relationship between environmental conditions, mosquito abundance and risk, which we know that at least for mosquito abundance and risk is not true but the implications are not mentioned and lead to confusion about the objectives/conclusions of the manuscript. I may be convinced that for looking at human risk areas, mosquito abundance doesn't matter because what the approach is telling us is that there are certain conditions that predispose risk. The consequence is that the mechanisms of risk is a black box, meaning we might be able to understand the context of risk, but not the dynamics (as mentioned in L122) that create risk.

The introduction doesn't put the objectives in context. A big part of the approach/conclusion seem to be identifying the environmental drivers of human risk but they are exposed as a list of factors that have been linked to mosquitoes and does not clearly identify what is the knowledge gap/novelty of the manuscript nor how CG will might impact these that could have had an impact on human WNV.

Finally, the broader implications of results/conclusions are not explained. Simply, why should we care that CG has been a driver of WNV.

Methods. These are not detailed enough. I appreciate that there is no development of the ML

algorithm (available tool in R). However, the explanatory variables and interactions among them should be clearly defined. What values were used for tree complexity, learning rate and how many trees? And were sensitivity analyses done on these? I'm assuming it was a Bernoulli BRT? How do you deal with temporal autocorrelation. What are the reanalysis datasets and how are these used? The human case data is from 2007 to 2019 but then predictions are done from climate data from 1900? Generally, not clear how time was dealt with. Explain why model was trained on present day data.

L252-253. I truly welcome the honesty! But doesn't this mean that your model is not predicting accurately enough? Models like this are usually difficult to validate but cases are happening where your model doesn't predict as suitable habitat.

I'm guessing that case reporting rates was not accounted for? This is probably increasing over time with awareness/knowledge of WNV. Can this have implications for the results?

Other suggestions:

Abstract: The MS does not investigate other drivers of WNV change so saying that CG is a primary driver seems a bit of an overstatement.

Figure 1: Is this training and testing data combined?

L162. Could this be due to other reasons such as underreporting rates, changes in DDT use etc?
L166: "climate change predominantly contributed" seems like a strong statement given that some plot so not show much difference and there are other potential factors. I have no doubts CG is accelerating risk but not sure results here are so strong.

Fig.2. I may be misunderstanding this figure but if panel a is counterfactual (no CG), and the maps show suitability, higher suitability appears to be on counterfactual, which contradicts the rest of the results.

L210-212. This should be discussed.

L223-225. there's 15 years between these. Can we claim they are linked?

L309: For absences, do you mean tested but not confirmed or all areas in Europe without a confirmed case were considered absences? All Europe is included, how does the choice of this absence area influence the results? E.g. if you took only southern countries?

A discussion of the implication that CG is increasing WNV seems to be missing.

Reviewer #3 (Remarks to the Author):

The authors present evidence of the direct contribution of climate change to the emergence and increased prevalence of West Nile virus in Europe. At a high level, quantifying the impacts of climate change is a key next step in science to begin to mitigate these impacts. This is original and impactful, particularly in direct comparison to the counter-factual non-climate change scenario. While it is well known that environmental factors such as temperature impact individual drivers of WNV (e.g. extrinsic incubation period, changing vector distributions), it is relatively novel to quantify the direct impact on risk based on data under climate change and counter-factual scenarios. The methods are robust and standard, the authors include relative impacts of the drivers and multiple useful views of the output under different climate scenarios.

A few questions and comments about the methods and presentation of the results:

1. Please talk more about using Presence/Absence versus incidence data for WNV in the model. The model just uses presence/absence but the authors present incidence as well. It seems like a combination of both in an analysis could be powerful, for example potentially showing that both presence of the pathogen and the impact in terms of total cases are changing. (or connecting suitability with incidence??)

2. It seems from my reading of the methods that the authors are assuming that places in Europe with no recorded cases are a pseudo-absence (WNV is likely under reported by quite a bit, and that under-reporting depends on the region, so is this biased?). How will considering under-reporting change the results? How pseudo-absences are used can impact the accuracy of models, sometimes significantly so more discussion on this would be helpful.

3. It would be helpful for the reader and probably more powerful in terms of communication to show the map of their algorithm's predicted niche for WNV under the observed climate versus the actual cases/"Presence" data (and perhaps an accompanying error map) so we can visually see model's performance. Also, including an accuracy metric (ideally normalized) in addition to AUC would make this more comparable in the future to other studies of the same region or similar studies of other regions.

4. It seems in the figures (e.g. Figure 2 a and 2 b) that the "counter factual" versus observed are mis-labeled. Either that or I'm completely missing the point since it looks like in row a, which is counter-factual with no climate change, the risk of WNV is higher by the end of the time period.

5. page 2, line 38 I think "sanitary" is the wrong word

6. Discussion, page 8, starting line 227, others have indeed seen that the factors that predict presence or overall incidence on a decadal scale are different from the factors that predict inter-annual differences in incidence (Gorris, Morgan E., et al. "Assessing the influence of climate on the spatial pattern of West Nile virus incidence in the United States." *Environmental Health Perspectives* 131.4 (2023): 047016. Note, this is from a previous postdoc of mine)

Reviewer 1

Summary: The authors present an analysis of the ecological niche of West Nile virus (WNV) in Europe and track changes in WNV suitability under historic climate projections. A state-of-the-art niche modeling approach (boosted regression trees) is used to associate climatic and land cover variables with the occurrence of human WNV cases as reported in The European Surveillance System (TESSy) database of the ECDC. The main research question addressed by this paper is to which extent the recent emergence of WNV in Europe can be attributed to climate change. The authors tackle an interesting and relevant research question. Overall, the work offers interesting insights into the climatic and land use drivers of WNV in Europe (most of which are known from previous literature), provides evidence for impact attribution, and uses advanced and robust methods. Given extensive previously published work on WNV ecological niche mapping already published (1,2) with similar and alternative methods as well as historical and future projections of WNV risk in Europe based on these (see below), the work does only provide minor novel insights and evidence for the field.

Main Strength: The authors use recently published historical and counterfactual climate datasets provided through ISMIP. A strength of the study is that they provide a direct comparison between model projections under the historical vs “no climate change” counterfactual scenarios. Moreover, the use and presentation of their results under 4 different climate models gives a good indication of sensitivity of their results to uncertainty in the climate datasets. The analysis of relative importance of different predictors, which is essential when using “black-box”-type machine learning models, is well presented.

Major Weakness: The added value and novelty of the study is not clear given other groups have previously published ecological niche modeling studies for WNV in Europe describing similar patterns and presenting similar findings (1). The most recent European Lancet Countdown report also uses the model by Farooq et al. to project historical trends in WNV risk for Europe (2). Although the Lancet Countdown indicator does not the climate change pre-industrial control counterfactual modelled scenarios in the previous work, it still provides a strong indication of to which extent changes in WNV risk can be attributed to climate change. The present work supports previous findings by approaching the topic from a slightly different angle but does not lead to major novel insights. Most of the relationships derived from the analysis of importance of the different predictors are known from previous studies in and outside of Europe. Although the new study provides some slightly alternative ranking of variable importance (such as highlighting the importance of humidity and summer vs spring temperatures (see Line 228-230) that further refine knowledge about the ecological niche of WNV in Europe.

[1] Farooq, Z., Rocklöv, J., Wallin, J., Abiri, N., Sewe, M. O., Sjödin, H., & Semenza, J. C. (2022). Artificial intelligence to predict West Nile virus outbreaks with eco-climatic drivers. *The Lancet Regional Health–Europe*, 17.

[2] van Daalen, K. R., Romanello, M., Rocklöv, J., Semenza, J. C., Tonne, C., Markandya, A., ... & Lowe, R. (2022). The 2022 Europe report of the Lancet Countdown on health and climate change: towards a climate resilient future. *The Lancet Public Health*, 7(11), e942-e965.

Answer: We appreciate the feedback of the Reviewer and the opportunity to clarify within the manuscript the novelty and added value of our study, which provides, for the first time, unequivocally scientific evidence of climate change impacts on the spatial expansion of West Nile virus (WNV) in Europe. While we acknowledge the existence of previous ecological niche modelling studies dedicated to WNV in the continent, the critical distinction of our research lies in the formal attribution of climate change impacts to an infectious disease. To our knowledge, there is no existing literature formally attributing and evaluating the impacts of climate change to an infectious disease in Europe or elsewhere.

While the Reviewer rightly mentioned the Lancet Countdown report's use of the model of Farooq and colleagues, it is essential to note that this indicator does not employ the climate change pre-industrial control

counterfactual model scenarios, a fundamental component of our study. Consequently, it cannot provide an indication as to whether changes in WNV risk can be directly attributed to climate change. Farooq *et al.* (2022) do not provide an attribution of observed changes in the occurrence of WNV cases to historical changes in climate. In contrast, their study only quantifies the sensitivity of the outcome (number of cases) to variations in climate. According to the IPCC definition, their study would therefore count as an ‘identification of weather sensitivity study’ but not as an ‘impact attribution study’ where ‘identification of weather sensitivity’ does not necessarily imply impact attribution.

Attribution studies establish a quantitative comparison between the current state of a system, influenced by climate change, and a counterfactual baseline representing the system’s condition in the absence of climate-related changes. This difference, i.e. the formally attributed impact of climate change, is the novel aspect of our research: it quantifies, for the first time, the extent to which climate change is responsible for variations in WNV risk.

From the definition given in section 16.2.1 of chapter 16 of the WG2 contribution to the IPCC AR6: *“Based on these general definitions and following the approach applied in WGII AR5 Chapter 18 (Cramer et al. 2014), we define an observed impact as the difference between the observed state of a natural, human or managed system and a counterfactual baseline that characterises the system’s state in the absence of changes in the climate-related systems, defined here as climate system including the ocean and the cryosphere as physical or chemical systems. **The difference between the observed and the counterfactual baseline state is considered the change in the natural, human or managed system that is attributed to the changes in the climate-related systems (impact attribution).** The counterfactual baseline may be stationary or may change over time, for example due to direct human influences such as changes in land use patterns and agricultural or water management affecting exposure and vulnerability to climate-related hazards (see Section 16.2.3 for methods on how to construct the counterfactual baseline). [...] **‘Identification of weather sensitivity’ refers to the attribution of the response of a system to fluctuations in weather and short-term changes in the climate-related systems including individual extreme weather events (e.g., a heatwave or storm surge).** Typical questions addressed include: ‘How much of the observed variability of crop yields is due to variations in weather conditions compared to contributions from management changes?’ (e.g., Ray et al. 2015, Müller et al. 2017) and ‘Can weather fluctuations explain part of the observed variability in annual national economic growth rates?’ (e.g., Burke et al. 2015). Identification of weather sensitivity may also address the effects of individual climate extremes, for example asking, ‘Was the observed outbreak of cholera triggered by an associated flood event?’ (e.g., Rinaldo et al. 2012, Moore et al. 2017b). [...] In this chapter, we explicitly distinguish between assessment statements related to ‘climate attribution’ (listed in Table SM16.21), ‘impact attribution’ (listed in Table SM16.22) and ‘identification of weather sensitivity’ (listed in Table SM16.23). The identification of ‘weather sensitivity’ does not necessarily imply that there also is an impact of long-term changes in the climate-related systems on the considered system.”*

Our study also includes the ‘identification of weather sensitivity’ step when building the empirical model testing the dependence of the WNV occurrence on seasonal air temperature, precipitation, and relative humidity. In this regard, our models are in fact comparable to models such as the one developed by Farooq and colleagues. However, knowing that these factors have a significant effect on WNV occurrence does only provide a hint that long-term climate changes may also have played a role in the long-term evolution of occurrences, it does not necessarily imply that observed long-term changes in occurrence are mainly attributed to long-term changes in climate. To this end, the effects of these long-term trends in climate have to be quantified and separated from the long-term effects of the other direct human influences such as land use and population changes. This requires forcing the model with the observed climate and a counterfactual climate where these long term changes have been removed. That has not been done in the study by Farooq *et al.* 2022 (that also is the main reference of van Daalen *et al.* 2022) or any other studies considered in the Lancet Countdown report. However, this step is critical as it turns out that an important part of the observed

trend is also due to population changes (see trend in the counterfactual simulations shown in Figure 3). We have now made these different points more explicit in the Abstract, Introduction, and Results sections.

In conclusion, the crux of our study's novelty and significance is the formal attribution of climate change impacts to WNV in Europe, a dimension hitherto unexplored in the literature. We believe that this innovative approach substantially contributes to the field of infectious disease epidemiology, climate change science, and public/one health. We appreciate your feedback and hope this response clarifies the unique contributions of our research.

Minor comments:

- I am wondering how hyperparameter tuning was done since it is not reported in the text. Boosted regression trees come with several hyperparameters that need to be adjusted to construct a model with good performance. Usually this is done by either evaluating the performance of the model for different hyperparameter settings on a separate validation set or by using cross-validation, whereas evaluation of the final model performance is done on a separate out-of-sample test dataset (which should not be seen by the model during training or during validation). Here the authors construct ten different tree models using cross-validation for evaluating the performance of their tree model. The important topic of hyperparameter tuning and evaluation of their model on a separate test dataset is missing.

Answer: We appreciate your insightful comment and concur with your observation regarding the importance of tuning BRT parameters for optimal performance. During the initiation phase of our study, we conducted an exploration of the impact of tree complexity and learning rate parameter values on the predictive performance of our models, as measured by the area under the receiver operating characteristic curve (AUC), which led to the conclusion that the choice of these parameter values does not impact their predictive performance.

In response to the Reviewer's comment, we now report a sensitivity analysis performed following the methodology outlined by Elith *et al.* (2008, PMID: 18397250). Specifically, we systematically varied tree complexity [1, 5, 10] and learning rate [0.05, 0.01, 0.005, 0.001, 0.0005] parameter values across ten replicates, each time computing the predictive performance of the resulting BRT model (see the new Table S3). Our findings indicate that the AUC exhibits minimal variation. Consequently, we are confident that the values chosen for the BRT parameters in our study (a tree complexity of 5 and a learning rate of 0.005) are appropriate and that their choice does not impact the predictive performance of our ecological niche models.

New Table S3. Sensitivity of the area under the receiving operator curve (AUC) support to the specification of the tree complexity and learning rate BRT parameters. For each combination of tested tree complexity and learning rate parameter values, we report the median AUC support [as well as the first and third quartiles] obtained across ten replicates based on the GSWP3-W5E5 reanalysis dataset.

Tree complexity	Learning rate				
	0.0005	0.001	0.005	0.01	0.05
1	0.84 [0.84-0.85]	0.86 [0.86-0.87]	0.88 [0.87-0.88]	0.88 [0.88-0.89]	0.88 [0.88-0.89]
5	0.89 [0.89-0.90]	0.90 [0.89-0.91]	0.90 [0.90-0.91]	0.90 [0.90-0.91]	-
10	0.91 [0.90-0.91]	0.91 [0.91-0.92]	0.91 [0.91-0.92]	0.91 [0.90-0.92]	-

- The manuscript is missing an explanation for why GSWP3-W5E5 was chosen as a reference climate dataset. What makes this climate model a reasonable choice for reporting/visualization results in the main text compared to the other three which are mainly reported in the supplementary material?

Answer: We appreciate the insightful comment of the Reviewer on the previous lack of clarity regarding the emphasis put in our manuscript on GSWP3-W5E5 results. We now explicitly acknowledge and detail in the text that each of the four factual climate datasets considered in our study has its individual strengths and weaknesses. The rationale behind the choice to first emphasise results based on GSWP3-W5E5 is rooted in the alignment of this reanalysis dataset with real-world conditions, particularly for the years coinciding with the time window of WNV case data obtained from the ECDC and on which our ecological niche models

were trained. W5E5 is considered the potentially closest approximation to reality as it is based on the latest version of the European Reanalysis (ERA5; Hersbach *et al.* 2020, doi: 10.1002/qj.3803) that was further corrected by observational data based on the WATCH Forcing Data methodology (Cucchi *et al.* 2020, doi: 10.5194/essd-12-2097-2020). To generate the counterfactuals, i.e. to construct a dataset that described a counterfactual world without long-term changes in climate since 1901, the W5E5 data had to be expanded backwards in time. For this extension, we first used version 1.09 of the Global Soil Wetness Project phase 3 (GSWP3) dataset (Kim 2017, doi: 10.20783/DIAS.501), bias-adjusted to W5E5 v2.0 in order to reduce discontinuities at the 1978-1979 transition (see Mengel *et al.* 2021, doi: 10.5194/gmd-14-5269-2021). As some variables in GSWP3 show discontinuities at every turn of the month that have been induced by the month-by-month bias adjustment applied in its creation (Rust *et al.* 2015, doi: 10.1175/JHM-D-14-0123.1), we additionally considered a backward-extension based on the Twentieth Century Reanalysis version 3 (20CRv3; Slivinski *et al.* 2019, doi: 10.1002/qj.3598; 2021, doi: 10.1175/JCLI-D-20-0505.1), interpolated to 0.5° and then bias-adjusted to W5E5 v2.0. Notably, 20CRv3-W5E5 data remain continuous at every turn of the month thanks to the application of ISIMIP3BASD v2.5 in running-window mode. Thus, the 20CRv3-W5E5 reanalysis dataset can be considered an update of GSWP3-W5E5. 20CRv3-ERA5 has then been introduced to allow for testing the sensitivity of the results to potential trend and variability artefacts in W5E5 that are related to the climatological infilling procedures used to deal with gaps in the station observations employed for the bias adjustment of ERA5 for the production of WFDE5 (for a detailed description of this caveat see <https://data.isimip.org/caveats/20/>). Finally, we have also considered the 'raw' 20CRv3 data interpolated to 0.5° but not bias-adjusted to any other dataset. This latter dataset had been included since it was generated with only one method and did not need to be combined with another dataset to fully cover the 20th century. Overall, considering all four reanalysis datasets (that have all been introduced in Mengel *et al.* 2021, doi: 10.5194/gmd-14-5269-2021) allows us to investigate the robustness of our findings to the choice of the reanalysis dataset. Those aspects are now explicitly clarified within the text.

- The discussion on the relationship between avian diversity and WNV risk (Line 87-90) does not reflect the complete state of evidence. The authors suggest the presence of a dilution effect based on one reference. However, WNV studies have detected both dilution and amplification effects potentially because WNV transmission potential depends on avian community composition.

Answer: We thank the Reviewer for their insightful comment, and we concur with the significance of incorporating discussion elements on both dilution and amplification effects. In accordance with the suggested enhancement, the related paragraph now reads as follows: “*Biodiversity loss can also promote transmission patterns as decreases in host community diversity could increase the vector-host encounter rate [39,40]. For example, a negative correlation has been found between bird diversity and WNV infection in vectors, at the regional scale in Missouri, and in humans, at the national scale in the USA [41]. On the other hand, some evidence also supports the assertion that avian biodiversity loss can be a contributing factor to the decline in mosquito infection rates and avian seroprevalence in Atlanta (Georgia, USA) 42].*”

- When plotting the response curves shown in Figure S2, I suggest the authors to report how values for the other input variables of the tree models were set. Is the shape of the response curves sensitive to the values selected for the other variables?

Answer: We thank the Reviewer for their comment. In the previous version of our manuscript and Figure S2 (now labelled S3), these response curves were obtained by computing the ecological suitability variation associated with one specific variable while all others were kept constant at their *median* values. To investigate the sensitivity of the response curves to the choice of fixing the other environmental variables at their median value, we have now also generated alternative response curves obtained when keeping the other environmental variables at their first and third quartile values (see the new Figure S3 below). Overall, while we detect an expected variation of the absolute predicted values (reported on the y-axes), our results clearly highlight that the response curve patterns remain globally unchanged across the three different procedures (i.e. fixing the other environmental variables at their median, first quartile or third quartile value).

[New] Figure S3. Responses curves of the ecological niche modelling. For each environmental factor, we report the response curve of each of the ten replicate boosted regression tree (BRT) models trained on present-day data retrieved from the ISIMIP3a reanalysis dataset GSWP3-W5E5. These response curves indicate the relationship between the environmental values and the response, i.e. the ecological suitability of WNV, and were obtained by computing the ecological suitability variation associated with one specific variable, while all others were kept constant at their median (red curves), first quartile (orange curves) or third quartile (blue curves) value.

Reviewer 2

The manuscript investigates the potential role of climate change on human WNV risk in Europe. They use BRT models on historical presence/absence data together with multiple environmental variables to then predict WNV ecological suitability based on climatic projections assuming CG or lack of (counterfactual). They found that the area ecologically suitable for WNV under a CG scenario (assumed current status) considerably increased during the last century compared to the no CG counterfactual.

The manuscript is very topical and well presented with clear (and pretty) figures. My main comments are about the lack of clarity in the objective/rationale/novelty and the description of the methods, which raises questions over the conclusions. I detail these and other minor comments below.

Answer: We hope that the revised version of our Abstract now makes the objective/rationale and novelty of our study more explicit: “West Nile virus (WNV) is an emerging mosquito-borne pathogen in Europe and represents a public health threat in previously non-affected European countries. While climate change has been cited as a potential driver of its spatial expansion on the continent, a formal evaluation of this causal relationship is lacking. Here, we investigate the extent to which WNV spatial expansion in Europe can be attributed to climate change while accounting for other direct human influences such as land use and human population changes. To this end, we trained ecological niche models to predict the risk of local WNV circulation leading to human cases to then unravel the isolated effect of climate change by comparing factual simulations to a counterfactual based on the same environmental changes but a counterfactual climate where long-term trends have been removed. Our findings demonstrate a notable increase in the area ecologically suitable for WNV circulation during the period 1901-2019, whereas this area remains

largely unchanged throughout the last century in a no-climate-change counterfactual. The human population at risk of exposure exhibits a drastic increase over the historical period and we show that this increase is partly due to historical changes in population density, but that climate change has also been a critical driver behind the heightened risk of WNV circulation in Europe.”

Our main objective is to evaluate the contribution of historical long-term changes in climate to the WNV spatial expansion in Europe. Previous studies could only demonstrate the sensitivity of the occurrence of infections to weather related indicators (and additional environmental factors). However, such a sensitivity does not necessarily imply a strong impact of long-term climate changes as this depends on the strengths of these historical changes and the interplay across different variables (e.g., air temperature, precipitation, relative humidity) as well as on the impacts induced by long-term changes in other drivers (see also our answer to the related comment of Reviewer #1). We have now highlighted these aspects more explicitly in the main text:

“WNV has circulated in Europe since the 1950s, but it is only in 1996 that a large human outbreak with 393 human cases was detected in Romania [27]. WNV is characterised by a high genetic diversity, with West Nile virus lineage 1 (WNV-1) and West Nile virus lineage 2 (WNV-2) mainly associated with disease in humans and animals. A phylogenetic analysis has shown that six lineages have so far been detected in Europe where WNV-2 had the largest number of sequences available, accounting for 82% of all WNV sequences detected in Europe so far, and the widest diffusion, since it has been found in at least 15 European countries [28]. Since its emergence on the continent, annual WNV outbreaks have been reported every summer in Mediterranean and central Europe [29]. Since its detection in the State of New York in 1999, WNV has also invaded the North American continent [30]. Between 1999 and 2021, the USA has reported >55,000 WNV cases, of which >27,500 led to a neuroinvasive disease and >2,500 to death (www.cdc.gov/westnile).

It was earlier demonstrated that the occurrence of the virus is linked to high temperatures in spring [33] and summer [34,35], drought in summer [34,35], and warm winters [35]. In addition, high spring and summer temperatures, lower water availability, and drier winter conditions were found to be main determinants of WNV occurrence across Europe [36]. While local WNV circulation in Europe has been shown to depend on weather conditions [5,31], so far, the effect of the historical long-term changes in climate on the occurrence of (human) infections on the continent has not been quantified. An overall high sensitivity to weather conditions does not necessarily imply a strong impact of long-term climate change, as this depends on the strengths of these long-term changes in climate, the interplay across the changes in different climate variables that may amplify or cancel out, and the impact of long-term changes in other environmental and/or anthropogenic drivers. Changes in land use could indeed also noticeably impact the circulation of such vector-borne pathogens [31]. For instance, irrigated croplands and highly fragmented forests are known to favour WNV outbreaks in Europe [34,35].”

To quantify the contribution of historical long-term changes in climate to the observed increase in WNV infections in Europe, we have used four newly available pairs of factual and counterfactual climate data where long-term trends in the factual data have been removed. Comparing (i) the factual simulations of suitable areas or number of people affected where our model is forced by the factual (observation-based) climate and direct human forcings (land use changes and changes in population patterns) to (ii) the counterfactual ‘no climate change’ simulations where the model is forced by the same information about the direct human forcings but the counterfactual climate allows for a quantification of the contribution of long-term climate change to historical trends in infections. Our results clearly show that it is not trivial to conclude that climate change is a major driver of the observed changes in infections just based on a ‘weather sensitivity study’. In contrast, our analysis of long-term trends demonstrates that direct human forcings (mainly changes in population patterns) were equally important in explaining long-term trends in infections (see trends in the counterfactual simulations reported in Figure 3).

Main comments:

Objective/motivation: The objective of the manuscript is not clear. Investigation of the contribution of CG to WNV emergence building on the IPCC framework (L103-111) is vague and confusing. In addition, 'emergence' seems to be used as a synonym to 'risk' or 'spatial expansion' as WNV has been in Europe for a while. What is the ISIMIP and the implications for WNV ecological suitability. How does the IPCC framework fit in all this?

Answer: We hope that the above-mentioned modifications of the text clarify the objective and novelty of our study. We have now also completed the section where the IPCC definition of 'impact attribution' is provided: *"In this context, the Working Group 2 of the Intergovernmental Panel on Climate Change (IPCC) devoted a section to the attribution of observed changes in human, natural and managed systems to climate change in its sixth assessment report (IPCC 2022, chapter 16.2.1 [43]). The framework outlined by the IPCC defines an "observed impact as the difference between the observed state of a natural, human or managed system and a counterfactual baseline that characterises the system's state in the absence of changes in the climate-related systems", where climate-related systems mean the climate-system itself including the ocean and the cryosphere (e.g., changes in sea level rise) as physical or chemical systems that are not relevant in this study. The IPCC then states that the "difference between the observed and the counterfactual baseline state is considered the change in the natural, human or managed system that is attributed to the changes in the climate-related systems (impact attribution)". "Changes in climate-related systems" explicitly mean "any observed long term-term change" no matter whether such a trend is induced by anthropogenic climate forcing or not [44]. The counterfactual impact baseline cannot be observed and thus needs to be modelled by an impact model. A precondition for impact attribution is that the impact model explains the observed phenomenon under consideration reasonably well given its drivers."*

Regarding the use of the word 'emergence', we agree with Reviewer's comment and have now replaced it by "spatial expansion" throughout the text.

Also, the approach does not consider vectors (nor are they ever discussed) so the assumption is that the presence of human cases implies presence of vectors. I agree this is necessarily true for VBDs, my concern is that it also assumes a linear relationship between environmental conditions, mosquito abundance and risk, which we know that at least for mosquito abundance and risk is not true but the implications are not mentioned and lead to confusion about the objectives/conclusions of the manuscript. I may be convinced that for looking at human risk areas, mosquito abundance doesn't matter because what the approach is telling us is that there are certain conditions that predispose risk. The consequence is that the mechanisms of risk are a black box, meaning we might be able to understand the context of risk, but not the dynamics (as mentioned in L122) that create risk.

Answer: We agree with the Reviewer that using the word 'dynamic' is indeed inappropriate in the context of our study. We have therefore decided to change this phrase that now reads as follows: *"We subsequently estimated the areas ecologically suitable for local WNV circulation leading to human cases since the beginning of the 20th century considering either the historical climate or its respective counterfactual."*

The vector distributions are indeed not explicitly incorporated into our ecological niche models for two reasons: (i) while some distribution data/estimates are available for the *Culex* species involved in WNV transmission cycle, there is no equivalent data encompassing the past century, which prevents to re-project the WNV ecological suitability from 1901 onward. (ii) Although one way to circumvent the first issue would be to train and project ecological niche models for those different *Culex* species in the past, such ecological niche models would themselves be based on the same or a very similar set of present-day environmental factors to the one also included in the set of variables used to train WNV ecological niche models. This would therefore lead to a circularity issue also making it more difficult to interpret the results (and in particular the relative influence value computed for each variable). Besides, as precisely mentioned by the Reviewer, we here aim to estimate the risk of local WNV circulation *given* local environmental conditions,

which implicitly involves a minimum and here unestimated level of mosquito vector abundance. Those aspects are now explicitly stated in the text.

The introduction doesn't put the objectives in context. A big part of the approach/conclusion seem to be identifying the environmental drivers of human risk but they are exposed as a list of factors that have been linked to mosquitoes and does not clearly identify what is the knowledge gap/novelty of the manuscript nor how CG will might impact these that could have had an impact on human WNV.

Answer: We thank the Reviewer for their feedback on this important point. As detailed above, we have now improved and completed the Abstract and Introduction sections accordingly.

Finally, the broader implications of results/conclusions are not explained. Simply, why should we care that CG has been a driver of WNV.

Answer: In the last paragraph of the Discussion section, we highlighted the importance of climate change as a critical public health challenge and the importance of considering it for future surveillance and interventions. As also stated in the text, we believe that our work allows filling an analytical gap between epidemiology and climate science by evaluating the causal chain between climate change and the spatial expansion of an arboviral disease such as the West Nile fever.

Methods. These are not detailed enough. I appreciate that there is no development of the ML algorithm (available tool in R). However, the explanatory variables and interactions among them should be clearly defined. What values were used for tree complexity, learning rate and how many trees? And were sensitivity analyses done on these? I'm assuming it was a Bernoulli BRT? How do you deal with temporal autocorrelation. What are the reanalysis datasets and how are these used? The human case data is from 2007 to 2019 but then predictions are done from climate data from 1900? Generally, not clear how time was dealt with. Explain why model was trained on present day data.

Answer: We thank the Reviewer for their comment and acknowledge that some methodological aspects were not sufficiently detailed in the previous version of our manuscript. The 'Ecological niche modelling' subsection of the Methods section has now been substantially detailed to provide to the reader more information on the specification of the BRT algorithm parameters and on the related sensitivity analysis (for the sensitivity analysis, we here also refer to our answer to the related comment of Reviewer #1). Furthermore, as answered to a related comment of Reviewer #1, we now also provide a far more detailed description of the different reanalysis datasets considered in our study.

As for the temporal correlation, we did not have to deal with such an issue in the specific methodological framework of our study. Indeed, our response variable is a binary variable defined as follows: the detection or not of at least one human case during the period 2007-2019 that is not labelled in the ECDC database as an 'imported case'. Consequently, we do not work on cumulative incidence data, which would then indeed require dealing with a temporal correlation within the response variable (e.g., by implementing a spatio-temporal cross-validation instead of a spatial cross-validation procedure). As now more explicitly stated in the text, due to the heterogeneous surveillance effort between countries or even across time for a given area, we indeed preferred to consider such a binary absence/presence variable rather than the reported incidence for the considered period of time. Otherwise, our analyses should then be based on the assumption that the surveillance effort was homogeneous across the study area and the considered time period, which is an incorrect assumption in the studied geographical framework.

L252-253. I truly welcome the honesty! But doesn't this mean that your model is not predicting accurately enough? Models like this are usually difficult to validate but cases are happening where your model doesn't predict as suitable habitat.

Answer: We thank the Reviewer for raising this pertinent concern. We take the opportunity to underline that our models were precisely not trained on environmental variables corresponding to the time window of

these latest reported human cases. Therefore, those latest case data do not constitute a potential 'validation' dataset, which is not something that is at our disposal in the present study even if we of course consider partitions of training and test datasets within our spatial cross-validation procedure. We fully agree with the Reviewer that it is important to be honest about the fact that our models did not predict that the areas of these new cases were particularly ecologically suitable for some local WNV circulation, even if this is not necessarily depicting a lack of prediction capacity for the reason outlined above.

I'm guessing that case reporting rates was not accounted for? This is probably increasing over time with awareness/knowledge of WNV. Can this have implications for the results?

Answer: We completely agree with the Reviewer that local reporting rates are likely increasing over time with awareness/knowledge of WNV. As outlined in our answer to one of the previous comments above, this is precisely one of the reasons why we did not consider available local incidence values as our response variable. We have now further detailed this aspect in the text.

Other suggestions:

Abstract: The MS does not investigate other drivers of WNV change so saying that CG is a primary driver seems a bit of an overstatement.

Answer: In our study, we also addressed changes in land use and human population as additional drivers of change. As illustrated in Figure 3, The repercussions of these factors have been quantified through a trend in the simulated counterfactual baseline. However, we of course agree with the Reviewer that only the climate change driver is here formally tested by comparing factual simulations to a counterfactual. We have now edited the Abstract to clarify this point for the readers.

Figure 1: Is this training and testing data combined?

Answer: Yes, in the sense that during the BRT algorithm, the dataset is partitioned into training and test datasets based on spatially-explicit folds according to a spatial cross-validation procedure. Specifically, in our study, we specified five distinct folds, and at each iteration of the BRT algorithm, a different fold is discarded from the training process to be used as the test dataset.

L162. Could this be due to other reasons such as underreporting rates, changes in DDT use etc?

Answer: Not for this statement in particular because we here specifically refer to the different trends observed in the reconstructions respectively driven by the factual and counterfactual data, whose only difference is the absence of climate change in the latter.

L166: "climate change predominantly contributed" seems like a strong statement given that some plot so not show much difference and there are other potential factors. I have no doubts CG is accelerating risk but not sure results here are so strong.

Answer: We agree with the comment addressed by the Reviewer and have edited that sentence as follows: "*These results indicate that climate change contributed to the escalation of the risk associated with West Nile virus circulation in Europe*" ('predominantly' has been removed).

Fig.2. I may be misunderstanding this figure but if panel a is counterfactual (no CG), and the maps show suitability, higher suitability appears to be on counterfactual, which contradicts the rest of the results.

Answer: We thank the Reviewer for having noticed this issue. The legend of Figure 2 has now been corrected.

L210-212. This should be discussed.

Answer: We agree with the Reviewer and have included an entire new paragraph dedicated to the differences observed between reanalysis datasets.

L223-225. there's 15 years between these. Can we claim they are linked?

Answer: We agree with the Reviewer that we cannot formally prove a link, but this is the reason why we here employ the verb 'coincide'.

L309: For absences, do you mean tested but not confirmed or all areas in Europe without a confirmed case were considered absences? All Europe is included, how does the choice of this absence area influence the results? E.g. if you took only southern countries?

Answer: In our analyses, all the European admin areas (displayed in Figure 3) for which we do not have at least one confirmed non-imported human case have been treated as pseudo-absences. Only sampling pseudo-absences in southern countries would indeed certainly impact our results but we wouldn't see the rationale of discarding from the pseudo-absence dataset northern NUTS3 polygons for which no non-imported human case has been reported, as this corresponds to a non-negligible source of information to train our ecological niche models.

Reviewer 3

The authors present evidence of the direct contribution of climate change to the emergence and increased prevalence of West Nile virus in Europe. At a high level, quantifying the impacts of climate change is a key next step in science to begin to mitigate these impacts. This is original and impactful, particularly in direct comparison to the counter-factual non-climate change scenario. While it is well known that environmental factors such as temperature impact individual drivers of WNV (e.g. extrinsic incubation period, changing vector distributions), it is relatively novel to quantify the direct impact on risk based on data under climate change and counter-factual scenarios. The methods are robust and standard, the authors include relative impacts of the drivers and multiple useful views of the output under different climate scenarios.

A few questions and comments about the methods and presentation of the results:

1. Please talk more about using Presence/Absence versus incidence data for WNV in the model. The model just uses presence/absence but the authors present incidence as well. It seems like a combination of both in an analysis could be powerful, for example potentially showing that both presence of the pathogen and the impact in terms of total cases are changing. (or connecting suitability with incidence??)

Answer: We agree that this is an important point, that we have now further clarified in the text. As answered to a related comment from Reviewer #2, it appeared more relevant to solely consider a binary absence/presence response variable rather than an incidence estimated from ECDC data because of the heterogeneous surveillance effort between countries. If instead based on incidence estimates, our analyses should then have been based on the assumption that the surveillance effort was homogeneous across the study area, which is an incorrect assumption in our geographical framework.

2. It seems from my reading of the methods that the authors are assuming that places in Europe with no recorded cases are a pseudo-absence (WNV is likely under reported by quite a bit, and that under-reporting depends on the region, so is this biased?). How will considering under-reporting change the results? How pseudo-absences are used can impact the accuracy of models, sometimes significantly so more discussion on this would be helpful.

Answer: We indeed treat as pseudo-absences the administrative areas with no reported human case (excluding the cases labelled as 'imported'); and with approximately 80% of West Nile virus infections being asymptomatic (Kramer *et al.* 2007, PMID: 17239804), we acknowledge that the underreporting likely has

an impact on our models and results. Yet, only treating administrative polygons with non-imported human cases as presence data allows focusing on the areas for which there is a confirmed and sufficient level of local circulation leading to human cases, whose ecological niche is precisely what we aim to model here. Furthermore, the main objective of the study is not to provide the most accurate and precise WNV niche modelling, which has been investigated in previous studies (Watts *et al.* 2021, PMID: 34485672; Farooq *et al.* 2022, PMID: 35373173; Sofia *et al.* 2022, PMID: 35889046) and would have required the inclusion of additional environmental predictors such as more detailed land use categories unavailable within the ISIMIP framework considered in our study. We thank the Reviewer for pointing out this aspect that is now more explicitly acknowledged in the text.

3. It would be helpful for the reader and probably more powerful in terms of communication to show the map of their algorithm's predicted niche for WNV under the observed climate versus the actual cases/"Presence" data (and perhaps an accompanying error map) so we can visually see model's performance. Also, including an accuracy metric (ideally normalized) in addition to AUC would make this more comparable in the future to other studies of the same region or similar studies of other regions.

Answer: We agree that it was not optimal to solely reporting/referring to AUC values, the use of the AUC metric having been repeatedly criticised in previous works due to its dependence on prevalence (i.e. the proportion of recorded sites where a given species is present; Lobo *et al.* 2008, doi: 10.1111/j.1466-8238.2007.00358.x; Jiménez-Valverde 2012, doi: 10.1111/j.1466-8238.2011.00683.x; 2014, doi: 10.1007/s10531-013-0606-1; Ghisbain *et al.* 2023, PMID: 37704726). In the revised version of our manuscript, we have now further assessed the predictive performance of our ecological niche models by computing a prevalence-pseudoabsence-calibrated Sørensen's index (SI_{ppc}) defined as follows (Sørensen 1948; Leroy *et al.* 2018, doi: 10.1111/jbi.13402; Ghisbain *et al.* 2023, PMID: 37704726):

$$SI_{ppc} = (2 \times TP) / (2 \times TP + x \times FP_{pa} + FN)$$

$$\text{where } x = (P / A) \times ((1 - prev_{sp}) / prev_{sp})$$

and

$$prev_{sp} = P / (P + A)$$

with TP corresponding to the number of true positives, FP_{pa} to the number of false positives associated with sampled pseudo-absence points, FN to the number of false negatives, P to the number of presence points, and A to the number of pseudo-absence points. This index has its lower limit at zero and its upper limit at one (interpreted as a maximal predictive performance). Because the computation of this index requires binary presence-absence data but ecological niche models instead return ecological suitability values ranging from '0' to '1', we performed an optimisation procedure similar to one adopted by Li & Guo (2013, doi: 10.1111/j.1600-0587.2013.07585.x) by varying the threshold values in the range [0, 1] with a 0.01 step increment, and eventually selecting the threshold value maximising the SI_{ppc} (Ghisbain *et al.* 2023, PMID: 37704726). We computed a SI_{ppc} for each of the ten independent ecological niche models (updated Table S1). As detailed in the updated Table S1, all averaged SI_{ppc} values are higher than 0.80 (>0.83). For each trained ecological niche model, we also report the evolution of the SI_{ppc} according to the ecological suitability threshold value ranging from 0 to 1 (new Figure S2).

Table S1. Predictive performance of ecological niche models. The table reports area under the curve (AUC) and prevalence-pseudoabsence-calibrated Sørensen index (SI_{ppc}) values computed for each ecological niche model trained in the present study with a spatial cross-validation approach. Specifically, we trained ten independent replicate boosted regression tree (BRT) models on present-day data retrieved from each ISIMIP3a reanalysis datasets considered in our study (GSWP3-W5E5, 20CRv3, 20CRv3-ERA5, and 20CRv3-W5E5). The SI_{ppc} were computed while performing an optimisation of the ecological suitability threshold in the range [0, 1] with a 0.01 step increment. This threshold value was used to generate binary versions of the ecological suitability maps necessary for the computation of this index, and we eventually selected the threshold value maximising the SI_{ppc} , which is here reported under parentheses.

	Area under the curve (AUC)				SI_{ppc} (and threshold value maximising SI_{ppc})			
	GSWP3-W5E5	20CRv3	20CRv3-ERA5	20CRv3-W5E5	GSWP3-W5E5	20CRv3	20CRv3-ERA5	20CRv3-W5E5
Replicate 1	0.87	0.86	0.86	0.85	0.86 (0.37)	0.90 (0.29)	0.91 (0.34)	0.87 (0.36)
Replicate 2	0.87	0.86	0.89	0.84	0.86 (0.37)	0.90 (0.31)	0.91 (0.37)	0.85 (0.35)

Replicate 3	0.85	0.80	0.83	0.85	0.85 (0.39)	0.85 (0.25)	0.88 (0.32)	0.86 (0.37)
Replicate 4	0.87	0.84	0.89	0.86	0.86 (0.37)	0.93 (0.32)	0.94 (0.38)	0.86 (0.34)
Replicate 5	0.84	0.82	0.87	0.84	0.86 (0.38)	0.89 (0.27)	0.93 (0.35)	0.87 (0.38)
Replicate 6	0.85	0.81	0.87	0.84	0.87 (0.38)	0.88 (0.29)	0.93 (0.39)	0.87 (0.37)
Replicate 7	0.85	0.81	0.87	0.85	0.83 (0.37)	0.92 (0.34)	0.93 (0.38)	0.84 (0.38)
Replicate 8	0.86	0.81	0.89	0.83	0.87 (0.38)	0.91 (0.32)	0.94 (0.46)	0.85 (0.39)
Replicate 9	0.83	0.82	0.88	0.86	0.84 (0.38)	0.89 (0.30)	0.92 (0.42)	0.86 (0.34)
Replicate 10	0.86	0.80	0.88	0.84	0.86 (0.35)	0.90 (0.31)	0.93 (0.41)	0.86 (0.37)

[New] Figure S2. Assessment of the predictive performance of ecological niche models based on the computation of the prevalence-pseudoabsence-calibrated Sørensen index (SI_{ppc}). Specifically, we computed the SI_{ppc} while performing an optimisation of the ecological suitability threshold in the range [0, 1] with a 0.01 step increment. This threshold value was used to generate binary versions of the ecological suitability maps necessary for the computation of this index, and we eventually selected the threshold value maximising the SI_{ppc} (see Table S1 for the optimised SI_{ppc} values and associated threshold).

4. It seems in the figures (e.g. Figure 2 a and 2 b) that the “counter factual” versus observed are mis-labeled. Either that or I’m completely missing the point since it looks like in row a, which is counter-factual with no climate change, the risk of WNV is higher by the end of the time period.

Answer: We thank the Reviewer for having noticed this issue. The legend of Figure 2 has now been corrected.

5. page 2, line 38 I think "sanitary" is the wrong word

Answer: We have now replaced “sanitary” by “public health”.

6. Discussion, page 8, starting line 227, others have indeed seen that the factors that predict presence or overall incidence on a decadal scale are different from the factors that predict inter-annual differences in incidence (Gorris, Morgan E., et al. "Assessing the influence of climate on the spatial pattern of West Nile virus incidence in the United States." *Environmental Health Perspectives* 131.4 (2023): 047016. Note, this is from a previous postdoc of mine).

Answer: We thank the Reviewer for their comment that has helped to complement the Discussion section of our study. We edited the related paragraph accordingly: “*Although anomalous higher temperature in spring was previously identified as a reliable early warning predictor for WNV human cases [33,36], we here find that the relative contributions of air temperature in summer and winter to the ecological suitability of WNV were higher than the one of spring temperatures. This discrepancy could be due to the definition of the response variable of the model: while other studies define it as incidence [33,56], we define it as a binary presence/absence variable, which is a different way to account for WNV circulation (see below for a motivation of that choice). Moreover, it has been shown that factors associated with WNV incidence in the USA on a 10-year scale could be different from those associated with inter-annual differences [57].*”

REVIEWER COMMENTS

Reviewer #1 (Remarks to the Author):

In the first answer to reviewer 1 in the rebuttal letter the authors state that the “for the first time find unequivocally scientific evidence of climate change impacts on the spatial expansion of WNV in Europe”. The highlight the formal attribution approach taken using the simulation of potential climate without climate change. However, it should be highlighted that these are simulations of a counterfactual and not the actual climate, but a simulation. A standard attribution approach in climatology would study the time trends and drivers, however, agreeably this is very challenging given the limited WNV data.

In this sense the analysis adds a little bit further insights from the trends already presented in the Lancet countdown going back to 1950. Particularly, by providing the trend prior 1950. However, the results instead depend on a climate model simulation of potential current climate given pre-historic boundary conditions. The argument is that the Farooq and the Lancet countdown studies “cannot provide evidence of the trends being attributable to climate change”. However, both the approach taken by the authors and the Farooq and Lancet countdown team uses relationships from fitting models on modern data (approx. the last decade) to attribute and predict trends and geographical patterns. So, the attribution in this sense rests on the same basis.

Further on, the approach taken by Farooq et al. is using a more advanced machine learning approach that has evolved beyond boosted regression tree (although admittedly they still share much in the base methodology both being boosted tree approaches). As described in the supplement of the Lancet countdown (where the indicator features with prediction of climate change induced changes since 1950), the indicator does rule out the influence of the other factors in the trend and geographical predictions of changes. Still, I agree with the authors that the pre-historic counterfactual prediction is perhaps a more formal attribution approach, but this is assuming the the pre-historic control and counterfactual climate model simulations are correct. This, however, induces other types of uncertainties which are not formally assessed. So, still in my view the added value is smaller compared to previous studies.

Further on the pseudo-absence selection may induce further bias as there is no formal way to define what is an absence, but this approach makes strong assumptions on where absence observations occur. It would be necessary to investigate this assumption further by selecting it PA by different strategies, for example, only using areas where presence never has been observed for PA, ie using a similar approach as by Brady et al.

Brady OJ, Gething PW, Bhatt S, Messina JP, Brownstein JS, Hoen AG, Moyes CL, Farlow AW, Scott TW, Hay SI. Refining the global spatial limits of dengue virus transmission by evidence-based consensus

Reviewer #3 (Remarks to the Author):

The authors answered my questions and provided thoughtful responses and additions to the paper. The revised paper is, in my opinion, ready for being published.

Reviewer 1

In the first answer to reviewer 1 in the rebuttal letter the authors state that the “for the first time find unequivocally scientific evidence of climate change impacts on the spatial expansion of WNV in Europe”. The highlight the formal attribution approach taken using the simulation of potential climate without climate change. However, it should be highlighted that these are simulations of a counterfactual and not the actual climate, but a simulation. A standard attribution approach in climatology would study the time trends and drivers, however, agreeably this is very challenging given the limited WNV data. In this sense the analysis adds a little bit further insights from the trends already presented in the Lancet countdown going back to 1950. Particularly, by providing the trend prior 1950. However, the results instead depend on a climate model simulation of potential current climate given pre-historic boundary conditions. The argument is that the Farooq and the Lancet countdown studies “cannot provide evidence of the trends being attributable to climate change”. However, both the approach taken by the authors and the Farooq and Lancet countdown team uses relationships from fitting models on modern data (approx. the last decade) to attribute and predict trends and geographical patterns. So, the attribution in this sense rests on the same basis. Further on, the approach taken by Farooq et al. is using a more advanced machine learning approach that has evolved beyond boosted regression tree (although admittedly they still share much in the base methodology both being boosted tree approaches). As described in the supplement of the Lancet countdown (where the indicator features with prediction of climate change induced changes since 1950), the indicator does rule out the influence of the other factors in the trend and geographical predictions of changes. Still, I agree with the authors that the pre-historic counterfactual prediction is perhaps a more formal attribution approach, but this is assuming the the pre-historic control and counterfactual climate model simulations are correct. This, however, induces other types of uncertainties which are not formally assessed. So, still in my view the added value is smaller compared to previous studies.

Answer: We thank the Reviewer for taking their time to carefully assess the revised version of our manuscript. We agree that our study is complementary to previous work by Farooq *et al.* and its further development in the *Lancet* countdown. Farooq *et al.* investigate the contribution of individual environmental drivers to WNV outbreaks in Europe from 2010 to 2019 using a machine learning classifier (XGBoost), and the *Lancet* countdown implements their approach to further explore the climate suitability of WNV between 1951 and 2020 using observed climate data from the ERA5-Land reanalysis. In this sense, none of them constitutes an attribution study, where an impact is investigated by removing one driver, analysing the difference, and providing the overall historic influence of climate. The novelty of our work lies in the fact that it constitutes **the first attribution** of West Nile virus expansion to climate change in Europe in line with the Intergovernmental Panel on Climate Change (IPCC) Working Group 2 (WG2) definition, therewith representing a relevant extension to the *Lancet* countdown findings. For reference, attribution in the sense of WG2 investigates whether a given system has changed beyond a specified counterfactual baseline that characterises a given behaviour in the absence of climate change. We understand the Reviewer’s concerns and hereafter discuss each of their other considerations.

First, recalling the IPCC Sixth Assessment Report (section 16.2.1, chapter 16), in which the WG2 defined an “*observed impact as the difference between the **observed state** of a natural, human or managed system and a **counterfactual baseline** that characterises the system’s state in the absence of changes in the climate-related systems*”, it is clear that the counterfactual baseline is not a direct observation but is obtained by detrending the observational (factual) climate data: the counterfactuals approximate a “no climate change” climate through the removal of the long-term trend related to global mean temperature change from the factual reanalysis datasets. This currently constitutes the gold standard approach when aiming to investigate the attribution of climate change to some specific observational changes. Early in our manuscript, we introduced the concept of the counterfactual baseline, but we acknowledge that the previous version of our manuscript could require an extra explanation of this concept. We therefore edited the related paragraph included in the Introduction section as follows: “*We use four observationally-based reanalysis climate datasets and their counterfactuals that were recently made available through the Inter-Sectoral*

Impact Model Intercomparison Project (ISIMIP). ISIMIP is dedicated to fostering impact attribution following the definition of the IPCC WG2 in an international modelling effort in its currently running ISIMIP3a phase. Specifically, the counterfactual climate data are obtained by detrending the observational (factual) climate data: the counterfactuals approximate a “no climate change” climate through the removal of the long-term trend related to global mean temperature change from the factual reanalysis datasets [44]. The resulting time series thus consist of stationary climate data obtained from observational daily data when removing the long-term trend while preserving the internal day-to-day variability [44].”

Second, we agree about the limitations behind the WNV data available and thus the difficulties of studying time series trends on WNV incidence and climate simultaneously. In that sense, we here propose to compare WNV ecological suitability under observational (factual) and counterfactual climate scenarios, as a feasible and efficient choice for attribution. Such data limitations are typical when studying climate impacts and the IPCC impact attribution framework is designed with such issues in mind.

Furthermore, we provide the quantification of the people at risk between scenarios, and acknowledge the contribution of other WNV drivers besides climate change, such as increases in human population and land use changes. Finally, we believe that we accounted for the potential biases of the climate data by considering four reanalysis datasets varying in their assumptions and presenting all the results. That being said, we believe our study evaluates coherently and comprehensively the attribution of WNV spatial expansion in Europe to climate change.

Further on the pseudo-absence selection may induce further bias as there is no formal way to define what is an absence, but this approach makes strong assumptions on where absence observations occur. It would be necessary to investigate this assumption further by selecting it PA by different strategies, for example, only using areas where presence never has been observed for PA, ie using a similar approach as by Brady et al. (Brady OJ, Gething PW, Bhatt S, Messina JP, Brownstein JS, Hoen AG, Moyes CL, Farlow AW, Scott TW, Hay SI. Refining the global spatial limits of dengue virus transmission by evidence-based consensus).

Answer: While we already follow the strategy consisting in only simulating and sampling pseudo-absences within areas where confirmed presence has never been reported, we agree with the Reviewer that the sampling of pseudo-absences can potentially impact our models and the subsequent comparisons between factual and counterfactual simulations. The approach proposed by Brady *et al.* is however not directly applicable as such in the setting of our study as we lack detailed surveillance effort information at the NUTS3 administrative level to implement the scoring system that these authors introduced. Yet, we now propose to investigate the robustness and sensitivity of our results to the sampling intensity of pseudo-absences across the study area. While the main analyses are based on pseudo-absences sampled across 100% of the administrative areas in which no presence has been confirmed, we now also consider alternative datasets of pseudo-absences sampled from only 50% and 75% of the administrative areas not associated with a presence record. In both cases, we re-trained 100 ecological niche models each based on a random selection of 50% or 75% of the original pseudo-absences considered in the main analyses. Based on these newly generated models, we then re-estimated the changes in the population at risk of exposure to WNV in Europe (new Figure S9). Overall, our results confirm that the inferred trends remain highly consistent with the results obtained when considering all potential pseudo-absences (Figure S9).

Figure S9. Investigations of the robustness and sensitivity of the estimates of human population at risk of exposure to the sampling intensity of pseudo-absences across the study area. Similar to Figure 3, we here report past changes in the population

at risk of exposure to WNV estimated for both a counterfactual baseline and the observed historical climate data retrieved from the GSWP3-W5E5 reanalysis dataset and while considering two different thresholds of ecological suitability (ES) above which an area was considered at risk (0.1 and 0.5). Contrary to Figure 3, the estimates reported here are however based on ecological niche models trained on random subsets of all available pseudo-absences (PAs) corresponding to all optimised NUTS3 administrative areas with zero confirmed non-imported human cases of WNV infection. Specifically, we considered both a subsampling of 50 and 75% of all pseudo-absence data and, for each percentage, re-trained 100 ecological ecological niche models each time based on a random subset of pseudo-absence data.

Reviewer 3

The authors answered my questions and provided thoughtful responses and additions to the paper. The revised paper is, in my opinion, ready for being published.

Answer: We would like to thank the Reviewer for the positive assessment of the revised version of our manuscript.